# Semantic Parsing by Large Language Models for Intricate Updating Strategies of Zero-Shot Dialogue State Tracking

**Yuxiang Wu**[1][*], **Guanting Dong**[1][*], **Weiran Xu**[1] [*]

[1]Beijing University of Posts and Telecommunications, Beijing, China

{yuxiangw,dongguanting,xuweiran}@bupt.edu.cn

## Abstract

Zero-shot Dialogue State Tracking (DST) addresses the challenge of acquiring and annotating task-oriented dialogues, which can be time-consuming and costly. However, DST extends beyond simple slot-filling and requires effective updating strategies for tracking dialogue state as conversations progress. In this paper, we propose ParsingDST, a new In-Context Learning (ICL) method, to introduce additional intricate updating strategies in zero-shot DST. Our approach reformulates the DST task by leveraging powerful Large Language Models (LLMs) and translating the original dialogue text to JSON through semantic parsing as an intermediate state. We also design a novel framework that includes more modules to ensure the effectiveness of updating strategies in the text-to-JSON process. Experimental results demonstrate that our approach outperforms existing zero-shot DST methods on MultiWOZ, exhibiting significant improvements in Joint Goal Accuracy (JGA) and slot accuracy compared to existing ICL methods.

## 1 Introduction

Dialogue State Tracking (DST) is crucial in Task-Oriented Dialogue (TOD) systems to understand and manage user intentions(Wu et al., 2019; Hosseini-Asl et al., 2020; Heck et al., 2020; Lee et al., 2021; Zhao et al., 2022). Collecting and annotating dialogue states at the turn level is challenging and expensive (Budzianowski et al., 2018), and commercial applications often need to expand the schema and include new domains. Therefore, it is vital to develop DST learning strategies that can adapt and scale effectively with minimal data.

Most of the existing fine-tuning methods for zero-shot DST have primarily focused on domain-transfer approaches (Hosseini-Asl et al., 2020; Lin et al., 2021b,a), which have not yielded satisfactory

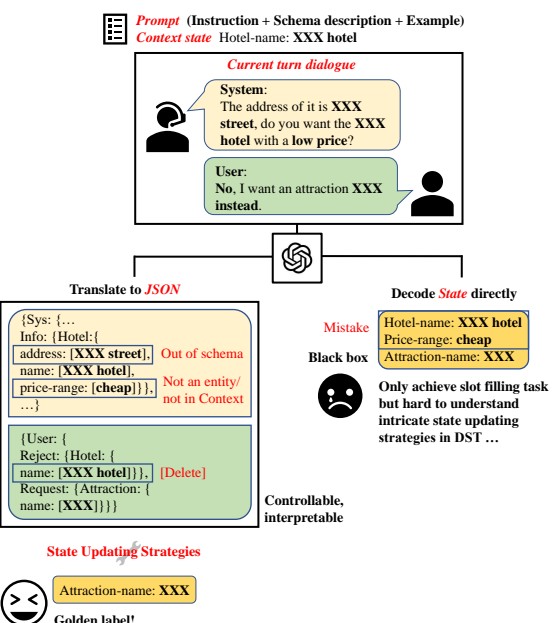

Figure 1: An example of the comparison between the previous ICL method and ours.

performance. Recent studies have showcased the impressive and adaptable expertise possessed by large language models (LLMs)(Raffel et al., 2020; Ouyang et al., 2022; Liu et al., 2023). However, with the discovery of the emergent ability of large language models in downstream tasks (Cobbe et al., 2021; Wei et al., 2022a,b), the In-Context Learning (ICL) method (Brown et al., 2020) has garnered more attention in DST research. ICL offers a more flexible and cost-effective approach as it eliminates the need for retraining when new slots or domains are introduced. Some recent studies have explored the effectiveness of ICL in DST (Hu et al., 2022; Madotto et al., 2021; Xie et al., 2022) such as the IC-DST method, which has demonstrated remarkable performance surpassing previous fine-tuning methods.

Nevertheless, all the previous methods directly generate the dialogue state or its change, which only achieves a simple slot-filling task that extracts the slot's value from dialogue but does not explain

---
[*]The first two authors contribute equally. Weiran Xu is the corresponding author.

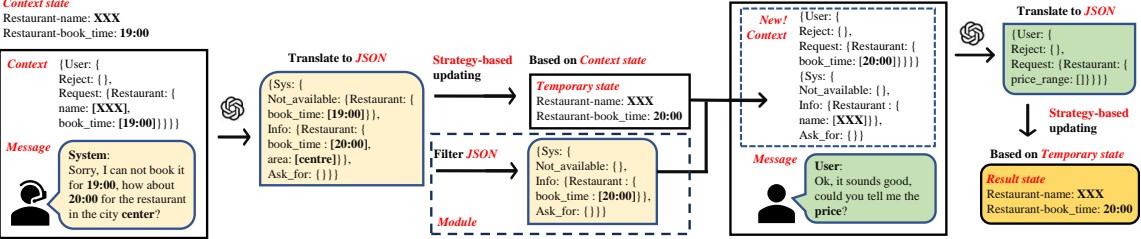

Figure 2: Our framework of ParsingDST that includes filter module.

how the state updating strategies work during DST which makes the DST process a black box. And we find the main obstacle in the current zero-shot DST method is that the large language models (LLMs)' understanding of updating strategies have not aligned with the DST task.

In this work, we proposed a new ICL method named ParsingDST for zero-shot DST. Specifically, we reformulate the DST task by semantic parsing that translates the **origin dialogue text into the designed JSON format (Text-to-JSON)** as the intermediate state through the LLMs The translated JSON data will include the information about the speaker, interaction action, domain, slot, and corresponding values in a formatted and structured manner, which enables the introduction of manual state updating strategies later.

We illustrated the significance of updating strategies by presenting an intricate scenario in Figure 1. Previous ICL approaches typically decode the dialogue to directly extract the values of all slots, such as "hotel-name", "price-range", and "attraction-name" in the example, without considering the updating strategies, which leads to errors in the result. In contrast, our method applies additional updating strategies after the text-to-JSON conversion. We delete non-entity slots like "price_range" when it appears in the system but not in the context, and we set the value of the "hotel name" to "[Delete]" when it is mentioned in the user's "reject" action, which makes our result aligns with the golden label.

Based on our observations, we noticed that when merging context, system, and user utterances as dialogue input like the previous methods, the LLMs will often exhibit confusion between these different types of information, which will invalidate the updating strategies. To address this, we also designed a companion framework with more process modules to ensure the effectiveness of updating strategies and conducted experiments to validate them.[1] We choose to test and compare ICL meth-

ods with the powerful LLMs by OpenAI[2]: gpt-3.5-turbo-0301 [3] and text-davinci-003 (Brown et al., 2020) on MultiWoz 2.1&2.4 dataset. :

In summary, our work makes the following contributions: (1) To our best knowledge, we are the first to apply text-to-JSON by semantic parsing as the transition between dialogue to state, making it possible to tool additional state updating strategies and other modules, making the process of DST more controllable and interpretable. (2) We proposed a novel framework with modules in the process of text-to-JSON to maintain the effectiveness of updating strategies. (3) We experiment with different ICL methods with various LLMs on MultiWOZ 2.1&2.4, proving our ICL method outperforms the previous in the same model and achieves a new state-of-art in zero-shot DST task.

## 2 DST System

### 2.1 Data format design

As we mentioned before, we translate the origin dialogue text to the JSON format which includes interactive actions. The domain, slot, and value in JSON are inferred from the utterance. And the formats of **User JSON** and **System JSON** are different. More details about JSON format and our prompt are in Appendix A.3.

### 2.2 Dialogue context representation

In order to maintain semantic consistency, we still use JSON format as context representation. When translating system's utterance, we put all domain-slot-value pairs from the context state into the "request" action of the data that is User JSON's format as context representation. Then When translating user's utterance, we merge the previous context representation and the System JSON as the new context representation.

[1]Codes in github.com/ToLightUpTheSky/ParsingDST

[2]More details about models in https://openai.com/
[3]Snapshot of gpt-3.5-turbo from March 1st, 2023, this model will not receive updates.

| MultiWoZ 2.1 | Attraction | Hotel | Restaurant | Taxi | Train | AVG |
|---|---|---|---|---|---|---|
| *Supervised fine-tuning method* | | | | | | |
| SimpleTOD++ (Hosseini-Asl et al., 2020) | 28.01 | 17.69 | 15.57 | 59.22 | 27.75 | 29.65 |
| T5DST + description (Lin et al., 2021b) | 33.09 | 21.21 | 21.65 | 64.62 | 35.42 | 35.20 |
| TransferQA (Lin et al., 2021a) | 31.25 | 22.72 | 26.28 | 61.87 | 36.72 | 35.77 |
| *Previous sota ICL method with different LLMs* | | | | | | |
| IC-DST Codex (deprecated) | 59.97 | 46.69 | 57.28 | 71.35 | 49.37 | 56.92 |
| IC-DST Text-davinci-003 | 50.69 | 38.55 | 43.98 | 71.25 | 45.99 | 50.09 |
| IC-DST Gpt-3.5-turbo-0301 | 59.32 | 40.20 | 46.50 | 68.32 | 52.87 | 53.13 |
| *Our method with different LLMs* | | | | | | |
| ParsingDST Text-davinci-003 | $64.60_{+13.91}$ | $39.92_{+1.37}$ | $62.55_{+18.57}$ | $\mathbf{80.45}_{+9.20}$ | $51.89_{+5.90}$ | $59.88_{+9.79}$ |
| ParsingDST Gpt-3.5-turbo-0301 | $\mathbf{64.95}_{+5.63}$ | $\mathbf{46.76}_{+6.56}$ | $67.04_{+20.54}$ | $80.26_{+11.94}$ | $\mathbf{62.78}_{+9.91}$ | $\mathbf{63.36}_{+10.23}$ |
| - filter module | $63.15_{+3.20}$ | $45.35_{+5.15}$ | $66.60_{+20.10}$ | $80.25_{+11.93}$ | $58.41_{+5.54}$ | $62.75_{+9.44}$ |
| - framework | $64.92_{+5.60}$ | $46.63_{+6.43}$ | $66.95_{+20.45}$ | $75.42_{+7.10}$ | $52.63_{-0.24}$ | $61.71_{+8.40}$ |
| **MultiWoZ 2.4** | **Attraction** | **Hotel** | **Restaurant** | **Taxi** | **Train** | **AVG** |
| IC-DST Gpt-3.5-turbo-0301 | 59.81 | 40.20 | 46.80 | 68.32 | 52.87 | 53.60 |
| ParsingDST Gpt-3.5-turbo-0301 | $\mathbf{65.63}_{+5.81}$ | $\mathbf{46.76}_{+6.56}$ | $67.67_{+20.87}$ | $80.58_{+12.26}$ | $62.59_{+9.72}$ | $\mathbf{64.65}_{+11.05}$ |

Table 1: Zero-shot per-domain JGA on MultiWOZ 2.1&2.4, we calculate the average of all per-domain JGA as the measure of overall performance, and we use subscripts to indicate the JGA changes of our methods compared to the previous sota ICL method under the same model.

## 2.3 Updating Strategies

Through the observation of the dataset, we have formulated some status update strategies, which are based on rulers and will influenced by the speaker, interactive action, and entity or not. As for slots in generated User JSON: (1) The slot in "reject" action will be deleted later; (2) The slot in "request" action will be updated. As for System JSON: (1) The slot in "ask_for" or "not_avaliable" actions wil be ignored, these actions are for reducing the mistakes of interaction classification in semantic parsing; (2) The slot in "info" action will be updated when it is an entity or it also in context state, but in other cases, we will ignore it .

## 2.4 Modules

The introduction of the new framework makes it possible to use additional modules to correct the generated previous content. In this paper, we tried a simple module that filters System JSON to make it only include the updated slot or entity slot in our framework.

## 2.5 Framework design

The motivation for designing this framework is to avoid harmful information from the history state or previous speaker being mixed into the JSON generated by the subsequent speaker and affecting the effectiveness of the update strategy. Our framework modifies the shared input-output method which processes all together. We make it asynchronous and add more modules in the pipeline: process the context, the system utterance, and the user utterance

dividedly. As shown in figure 2. The details of the steps in our framework pipeline are as follows:

Step1. In the $t$ turn of dialogue, convert the context state $S_{t-1}$ to JSON format $C_t$ as context representation (mentioned in section 2.2) in equation (1), then merge it with the system utterance $A_{utt}^t$ as the dialogue input for LLM, generate the System JSON $A_{JSON}^t$ in equation (2):

$$C_t = State2JSON(S_{t-1}) \tag{1}$$

$$A_{JSON}^t = Trans2JSON(C^t, A_{utt}^t) \tag{2}$$

Step2. Then the context state is updated by updating strategies and the System JSON and saved as the temporary state $S_{temp}^t$ as in equation (3):

$$S_{temp}^t = Update(S_{t-1}, A_{JSON}^t) \tag{3}$$

Step3. Equation (4) will filter the generated System JSON with the module mentioned in section 2.4:

$$A_{JSON}^t = Filter(A_{JSON}^t) \tag{4}$$

Step4. Equation (5) combines context JSON and filtered System JSON as the new context representation, then equation (6) inputs the new context with user's utterance $U_{utt}^t$ together and generate User JSON $U_{JSON}^t$ with LLM:

$$C_t = C_t + A_{JSON}^t \tag{5}$$

$$U_{JSON}^t = Trans2JSON(C_t, U_{utt}^t) \tag{6}$$

Step5. Finally, the temporary state is updated by the updating strategies and the User JSON to get our dialogue state of this turn $S_t$ in quation (7):

$$S_t = Update(S_{temp}^t, U_{JSON}^t) \tag{7}$$

If it is the first turn, the process will start from step4 and $A_{JSON}^t$ will be empty.

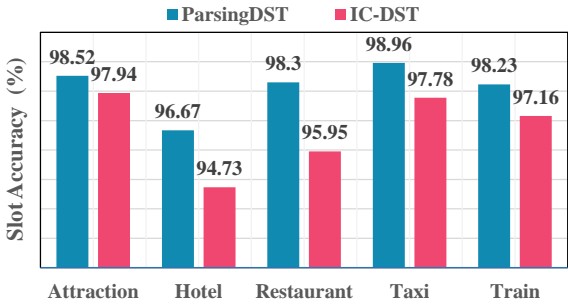

Figure 3: The average slot accuracy in 5 different domains of MultiWOZ 2.1 with GPT -3.5 model.

# 3 Experiment

## 3.1 Dataset, Metric, and Evaluation

MultiWOZ (Budzianowski et al., 2018) is a multi-domain human-human written dialogue dataset, the labels and utterances have been refined in subsequent versions, e.g., MultiWOZ 2.1 (Eric et al., 2019) and MultiWOZ2.4 (Ye et al., 2021) is a cleaner version. We use Joint Goal Accuracy (JGA) which is the average accuracy of predicting all slot assignments for a given service in a turn correctly to evaluate the main results of models. Regarding text preprocessing and labeling, we followed most of the previous work of IC-DST (Hu et al., 2022) on preprocessing and labeling data. The only change is that we renamed some names of the slots and domains. More details about baselines and zero-shot settings are in Appendix A.1 and Appendix A.2.

## 3.2 Results and Analysis

**Main Result.** Table 1 shows the zero-shot per-domain JGA result on MultiWOZ 2.1, and AVG JGA as the measure of overall performance. Because IC-DST's best performance model Codex-Davinci (Chen et al., 2021) has been deprecated by openAI, we supplement two experiment results with GPT-3.5 and Text-davinci models. We can find from the result that the supervised fine-tuning method is far inferior to the ICL method. And no matter in which domain, our method always outperforms the previous sota ICL method IC-DST by a large-scale margin in the same model. For example, using the same GPT-3.5 model, we achieved a 20.54% increase in JGA in the restaurant domain. Although the supplemented experiments of IC-DST show worse performance compared to the Codex model, our ParsingDST method still outperforms the IC-DST Codex in all five domains with the same model of supplemented experiments, which shows the strong ability of our method and the great

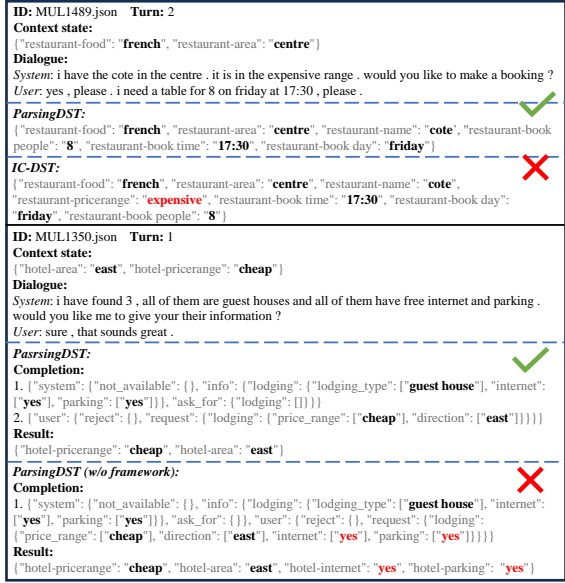

Figure 4: Two representative samples in the test set of MultiWOZ 2.1.

influence of updating strategies in zero-shot DST.

**Ablation Studies.** We conduct an ablation study to better prove the effectiveness of our modules and new framework. In the situation without the framework, we follow the traditional setting to merge context, user, and system utterance as dialogue input, then output JSON all at once. As shown in Table 1, the overall performance degrades whether the filter module or our framework is discarded. However, we observed that the JGA in certain domains remained similar to the previous performance during our ablation studies, we believe that is because our modules and updating strategies still need to be optimized.

**Slot Accuracy Analysis.** Figure 3 shows the slot accuracy of models using PasingDST and IC-DST with the Gpt-3.5 model. It can be seen that our method achieves better results on all 5 domains compared to the IC-DST (e.g., 2.88% improvement in the restaurant domain), which further proves the effectiveness of our method. We speculate that it is crucial for DST to introduce an appropriate dialog state update strategy into the model.

**Case Studies.** To further illustrate the effectiveness of our framework, Figure 4 shows two representative samples of different methods' predictions. In the first case, the IC-DST method makes a wrong prediction that includes the non-entity slot "pricerange" from system utterance that does not appear in context, however, our method can handle the situation well because of the introduction of updating strategies. In the second case, parsing-

DST makes a wrong prediction when removing its framework, because its User JSON included slots "internet" and "parking" from system utterance, which makes the updating strategies invalid because these slots will be updated rather than be ignored. In our method, we can process System JSON and context JSON with modules that are included in the framework before the harmful information influences the latter User JSON.

## 4 Conclusion

In this paper, we reformulate DST to semantic parsing with LLMs that translate the dialogue text into JSON as the intermediate state, which enables us to introduce updating strategies and makes DST process more controllable and interpretable. Furthermore, we present a novel framework that includes more modules within the text-to-JSON conversion process to ensure the effectiveness of updating strategies. Through experiments on the MultiWOZ 2.1&2.4 dataset, our system achieves sota performance in zero-shot settings. Our analyses showed that our method surpasses the previous ICL method and the innovative modules and framework benefit the overall performance. In future work, we plan to explore and evaluate more variety of updating strategies and modules to further enhance our framework and we will test our method in more open-source LLMs like Llama (Touvron et al., 2023).

## Limitations

This work has two main limitations: (1) The performance of our framework is highly dependent on the inference language model, which may limit the framework's usage. For example, it depends on the ability of LLMs that are pre-trained in JSON and natural language data and can understand both. (2) Lack the detailed comparison of various updating strategies and modules in the framework. We will design and test more kinds of updating strategies and modules for our framework in future work.

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

# A Appendix

## A.1 Baseline

**SimpleTOD (Hosseini-Asl et al., 2020)** Successfully leverage the pretrained language model GPT-2 for the end-to-end TOD modeling in the unified way.

**T5DST (Lin et al., 2021b)** A slot description enhanced approach for zero-shot & few-shot cross-domain DST based on T5.

**TransferQA (Lin et al., 2021a)** Reformulated DST as QA problem. The model is pre-trained with a large amount of QA data. At inference time, the model predicts slot values by taking synthesized extractive questions as input

**IC-DST (Hu et al., 2022)** An in-context learning method for zero-shot & few-shot DST by text-to-SQL with LLMs. It's the previous state-of-art method that outperforms the supervised fine-tuning methods in zero-shot DST.

## A.2 Zero-shot setting of ICL frameworks

There are no labeled examples to retrieve, but formatting examples are included, following previous zero-shot learning work. (Wang et al., 2022), and we set the temperature parameter of LLMs to 0 for a stable result.

## A.3 Prompt design

Below are the template of the user and system's prompt in zero-shot settings. The last example is the test instance that needs to be completed. We find the example is more useful than the task-description prompt in the in-context learning, so we use a simple example that includes all slots of the domain to map the relationship between slot and value in the dialogue rather than directly describe the slot and its value. Furthermore, for LLMs to understand the meaning of action in JSON format, we also formulate some examples to demonstrate some interaction typical scenarios.

**System prompt template**

translate system message to JSON:

data format of JSON:

input message:
system: "..."
output JSON:
{"system": {"not_available": {domain: {slot: [value]}}, "info": {domain: {slot: [value]}}, "ask_for": {domain: [slot]}}}
[END]

example:

input message:
system: "the booking for restaurant at 10:00 on sunday was successful"
output JSON:
{"system": {"not_available": {}, "info": {"restaurant": {"clock_book": ["10:00"], "week_day": ["sunday"]}}, "ask_for": {}}}
[END]

example:
input message:
system: "it is a chinese restaurant in the centre"
output JSON:
{"system": {"not_available": {}, "info": {"cuisine": ["chinese"], "direction": ["centre"]}, "ask_for": {}}}
[END]

example:
input message:
system: "how about abc restaurant in the city centre"
output JSON:
{"system": {"not_available": {}, "info": {"restaurant": {"full_name": ["abc restaurant"], "direction": ["centre"]}}, "ask_for": {}}}
[END]

example:
input message:
system: "how about the part of the area and food type for the restaurant"
output JSON:
{"system": {"not_available": {}, "info": {}, "ask_for": {"restaurant": ["direction", "cuisine"]}}}
[END]

example:
input message:
system: "do you need certain price range or part of area for restaurant"
output JSON:
{"system": {"not_available": {}, "info": {}, "ask_for": {"restaurant": ["price_range", "direction"]}}}
[END]

example:
input message:
system: "sorry i can not book restaurant nusa for you . i can only find nandos"
output JSON:
{"system": {"not_available": {"restaurant": {"full_name": ["nusha"]}}, "info": {"restaurant": {"full_name": ["nandos"]}}, "ask_for": {}}}
[END]

[DM]
[ST]
[KW]

example:
context:
[PREDIC]
input message:
[DIALOG]
output JSON:

## User prompt template

translate user message to JSON:

data format of JSON:

input message:
user: "..."
output JSON:
{"user": {"reject": {domain: [slot]}, "request": {domain: {slot: [value]}}}}
[END]

example: input message:
user: "i want a place to eat . in the city centre . with cheap price"
output JSON:
{"user": {"reject": {}, "request": {"restaurant": {"direction": ["centre"], "price_range": ["cheap"]}}}}
[END]

example:
input message:
user: "no particular food type"
output JSON:
{"user": {"reject": {}, "request": {"restaurant": {"cuisine": ["any"]}}}}
[END]

example:
{"system": {"not_avaliable": {}, "info": {}, "ask_for": {"restaurant": ["price_range", "cuisine"]}}}

input message: user: "no , i am not picky as long as it book for 4 on sunday"
output JSON:
{"user": {"reject": {}, "request": {"restaurant": {"price_range": ["any"], "cuisine": ["any"], "num_people": ["4"], "week_day": ["sunday"]}}}}
[END]

example:
input message:
user: "i want to be in the east of town . can i get their phone number and address please"
output JSON:
{"user": {"reject": {}, "request": {"restaurant": {"direction": ["east"], "phone_number": [], "address": []}}}}
[END]

example:
input message:
user: "nusha is not a restaurant but an attraction"
output JSON:
{"user": {"reject": {"restaurant": ["full_name"]}, "request": {"attraction": {"full_name": ["nusha"]}}}}
[END]

[DM]
[EXM]
[ST]
[KW]

example:
context:
[PREDIC]
input message:
[DIALOG]
output JSON:

## Special tokens in prompt

The prompt has some special tokens which will be replace by something else in the data-bulding process:

"[DM]" is the domain list.

"[EXM]" is the domain example which is a user utterance that includes all slots of the domain and its translated JSON.

"[ST]" is the slot list of the domain.

"[KW]" is the possible choice of some slot.

"[PREDIC]" is the context JSON.

"[DIALOG]" is the utterance that should be translated to JSON.

"[END]" is the stop token.