# OpenReview forum: "Semantic Parsing by Large Language Models for Intricate Updating Strategies of Zero-Shot Dialogue State Tracking"
_EMNLP/2023/Conference — EMNLP 2023 Findings_

### Official Review · Reviewer_Qxvg · 2023-07-31

**Soundness:** 2

**Excitement:**

3: Ambivalent: It has merits (e.g., it reports state-of-the-art results, the idea is nice), but there are key weaknesses (e.g., it describes incremental work), and it can significantly benefit from another round of revision. However, I won't object to accepting it if my co-reviewers champion it.

**Paper Topic And Main Contributions:**

The paper proposes a novel approach to use large language models for dialogue state tracking, in the in-context learning framework. Rather than directly proposing the target belief state, they propose to generate a structured representation that accounts for the user's acceptance or rejection of a suggestion, and accounts for whether the assistant was making a suggestion or asking for clarification. The formal representation is updated throughout the dialogue with a combination of a LLM prediction and a rule-based state update based on the previous formal state.

**Questions For The Authors:**

Can you elaborate on what exactly is the framework, and what it means to evaluate your approach with and without the framework?

**Reasons To Accept:**

The paper achieves a significant improvement over the state of the art zero-shot performance on the well-known MultiWOZ 2.1 benchmark.

**Reasons To Reject:**

The paper only evaluates on MultiWOZ 2.1. It is well-known that MultiWOZ, and MultiWOZ 2.1 specifically, is plagued by annotation issues, which subsequent revisions have attempted to address (See for example Ye et al https://arxiv.org/abs/2104.00773, Campagna et al https://arxiv.org/abs/2009.07968 ). In particular, previous work has noted that MultiWOZ 2.1 does not have a well-defined and consistently-followed rule of whether a mentioned slot should be included in the "gold" belief state.
Given the description of the errors provided by the authors themselves in the introduction, the most likely explanation of the improvement is not that the model is measurably better, but that the update strategy tracks better the quirks of MultiWOZ's poor and inconsistent annotation scheme.

**Reproducibility:**

3: Could reproduce the results with some difficulty. The settings of parameters are underspecified or subjectively determined; the training/evaluation data are not widely available.

**Reviewer Confidence:**

5: Positive that my evaluation is correct. I read the paper very carefully and I am very familiar with related work.

**Typos Grammar Style And Presentation Improvements:**

Section 2.4 was very hard to follow. It would help to clarify which functions are LLMs and which functions are rule-based.

---

> ### Author Rebuttal · Authors · 2023-08-24
>
> Thank you for your insightful suggestions and questions. Below is our detailed response:
>
> **Q1**: The paper only evaluates on MultiWOZ 2.1.
>
> **A1**: Hello, we used the settings that performed best on Moz 2.1 and supplemented our method with JGA results on Moz 2.4 (the performance improvements compared to IC-DST are shown in parentheses below):
>
> Per-domain JGA of **IC-DST Gpt-3.5-turbo-0301** and **ParsingDST Gpt-3.5-turbo-0301** in Moz 2.4:
>
> |Model| attraction | hotel | restaurant | taxi | train | avg |
> | :---- | :---- | :----: | :----: | :----: | :----: | :----: |
> |IC-DST Gpt-3.5-turbo| 59.81 | 40.20 | 46.80 | 68.32 | 52.87 | 53.60 |
> |ParsingDST Gpt-3.5-turbo| 65.63(+5.81) | 46.76 (+6.56) | 67.67(+20.87) | 80.58(+12.26) | 62.59(+9.72) | 64.65(+11.05)|
>
> As we can see, our method still achieves very satisfactory results and has significantly improved compared to the IC-DST method. We will supplement our results on MultiWOZ 2.4 in the future to demonstrate our generalization ability.
>
> **Q2**: Can you elaborate on what exactly is the framework, and what it means to evaluate your approach with and without the framework?
>
> **A2**: As mentioned in the introduction, after receiving the JSON output of the LLM (mentioned in section 2.1), we refer to the information in this result and update the current dialogue state using the updating strategy (mentioned in section 2.3). The updating strategies adopted for the JSON generated by different speakers are also different.
>
> Therefore, the motivation for designing this framework is to avoid harmful information from the history state or previous speaker being mixed into the JSON generated by the subsequent speaker and affecting the effectiveness of the update strategy.
>
> Usually, when we obtain the dialogue state, we input the history state and messages from both the user and the system to LLM and obtain the outputs of both together. Our framework modifies this shared input-output to be asynchronous. In each turn, we first input the history state and message of the previous speaker to LLM to obtain the JSON result, and then use additional modules (such as a simple filter module mentioned in section 2.3 of the paper to process the JSON from the system) to make rule-based modifications before using the update strategy to obtain a temporary dialogue state, which is used to guide the LLM's generation of the JSON result from the subsequent speaker. The update strategy is then used again to obtain the final dialogue state of the current turn. This asynchronous input-output framework avoids harmful information from previous context or speaker affecting the content generated later in the decoding process with more processing module and ensures the effectiveness of the update strategy.
>
> In the absence of a framework, it means that the messages from both the user and the system in the turn are jointly inputted to LLM to obtain the JSON output of both at once. Afterwards, the final dialogue state of the current turn is obtained by accepting rule-based updating strategy processing. However, because the updating strategies adopted by different speakers are different, in the decoding process, the harmful information of the history state or the previous speaker generated may impact the information of the later speaker (such as confusing non-entity slot values from the system into the generated user JSON), which affects the results obtained through the updating strategy.
>
> We have already described the part about our framework in the Introduction section and Framework design section (Section 2.4), and the above content is a more detailed explanation. We hope it can help you further understand our framework more easily.
>
> **Q3**: Section 2.4 was very hard to follow. It would help to clarify which functions are LLMs and which functions are rule-based.
>
> **A3**: Our section 2.4 is a further explanation of the framework diagram in Figure 2, where the order corresponds to the process of gradually obtaining the final dialogue state from left to right in the figure. The function state2JSON for converting slot-value type state to JSON format context is mentioned in section 2.2, and the rule-based update and filter functions are mentioned in the paper as corresponding respectively to the updating strategies and modules in section 2.3. If other functions need to use LLM, we have explained it in the paper such as Trans2JSON. Thank you for your feedback. We will consider adding references to previous sections in section 2.4 to make the article easier to understand, as well as adding a data preprocessing chapter in the appendix.

---

### Official Review · Reviewer_Zu2U · 2023-08-05

**Soundness:** 3

**Excitement:**

3: Ambivalent: It has merits (e.g., it reports state-of-the-art results, the idea is nice), but there are key weaknesses (e.g., it describes incremental work), and it can significantly benefit from another round of revision. However, I won't object to accepting it if my co-reviewers champion it.

**Paper Topic And Main Contributions:**

This paper introduces ParsingDST, a new In-Context Learning (ICL) method for zero-shot Dialogue State Tracking (DST) that reformulates DST as a text-to-JSON task. Previous ICL approaches typically decode the dialogue to directly extract the values of all slots without considering the updating strategies. In contrast, this method applies additional updating strategies after the text-to-JSON conversion. This allows for manual updates, too.

The main contributions are: reformulating DST as a text-to-JSON task and proposing new state-updating strategies (given the extracted JSON).

**Questions For The Authors:**

- How did you preprocess the text?
- How did you preprocess the labels?

Bonus:
- Nothing wrong with using closed-source models; would be nice (i.e., useful for practitioners in this field) to have results with open-source LLMs.

**Reasons To Accept:**

- Interesting and useful method

**Reasons To Reject:**

- Only evaluated on the 2.1 version of MultiWoz which is known to have noisy annotations. The paper would benefit from evaluation of other versions (e.g, 2.2 and 2.4)
- The data preprocessing and evaluation are not explained in depth. Since results are very much dependent on these factors, the paper needs to add more details about it to improve its soundness (happy to raise the soundness score once addressed).

**Reproducibility:**

4: Could mostly reproduce the results, but there may be some variation because of sample variance or minor variations in their interpretation of the protocol or method.

**Reviewer Confidence:**

4: Quite sure. I tried to check the important points carefully. It's unlikely, though conceivable, that I missed something that should affect my ratings.

**Typos Grammar Style And Presentation Improvements:**

Overall the manuscript is not clearly written in some parts. For example:

- The updating strategy section is a bit unclear. Would be good to expand on that in the camera-ready if the paper is accepted. Again, I would also stress the data preprocessing part: I would strongly suggest adding a section in the appendix.

- The grammar is a bit problematic in some points with obvious mistakes (e.g., appendix section title "Explains" to mean "More details"?) and some parts even if not wrong are not written. The paper would need a thorough revision of the exposition.

---

> ### Author Rebuttal · Authors · 2023-08-24
>
> Thank you for your insightful suggestions and questions. Below is our detailed response:
>
> **Q1**: Only evaluated on the 2.1 version of MultiWoz which is known to have noisy annotations.
>
> **A1**: Hello, we used the settings that performed best on Moz 2.1 and supplemented our method with JGA results on Moz 2.4 (the performance improvements compared to IC-DST are shown in parentheses below):
>
> Per-domain JGA of **IC-DST Gpt-3.5-turbo-0301** and **ParsingDST Gpt-3.5-turbo-0301** in Moz 2.4:
>
> |Model| attraction | hotel | restaurant | taxi | train | avg |
> | :---- | :---- | :----: | :----: | :----: | :----: | :----: |
> |IC-DST Gpt-3.5-turbo| 59.81 | 40.20 | 46.80 | 68.32 | 52.87 | 53.60 |
> |ParsingDST Gpt-3.5-turbo| 65.63(+5.81) | 46.76 (+6.56) | 67.67(+20.87) | 80.58(+12.26) | 62.59(+9.72) | 64.65(+11.05)|
>
>
> As we can see, our method still achieves very satisfactory results and has significantly improved compared to the IC-DST method. We will supplement our results on MultiWOZ 2.4 in the future to demonstrate our generalization ability.
>
> **Q2**: Nothing wrong with using closed-source models; would be nice (i.e., useful for practitioners in this field) to have results with open-source LLMs.
>
> **A2**: Thank you for your affirmation. We believe your opinion is very valuable. As we mentioned in the Limitations section, this in-context method relies on the capabilities of LLM, which are trained in JSON format data and natural language. In future work, we will further demonstrate the effectiveness of our method using models such as Llama and ChatGLM etc.
>
> **Q3**: The updating strategy section is a bit unclear. Would be good to expand on that in the camera-ready if the paper is accepted. Again, I would also stress the data preprocessing part: I would strongly suggest adding a section in the appendix.
>
> **A3**: We apologize for any confusion caused by our presentation. In section 2.3, we mentioned the update strategy for representing the current slot-value form of the state based on the content of the JSON generated by LLM. This is a rule-based processing strategy. We used the if-else expression to make the expression of the rules more concise and accurate, in accordance with our code-writing logic. The JSON forms generated by user and system utterances were mentioned in section 2.1. We will sort out the expression in the main text and add a detailed data processing section in the appendix to make the relevant parts easier to understand.
>
> **Q4**: The grammar is a bit problematic in some points with obvious mistakes (e.g., appendix section title "Explains" to mean "More details"?) and some parts even if not wrong are not written. The paper would need a thorough revision of the exposition.
>
> **A4**: Alright, we will double-check our grammar expression. However, we believe that these minor differences in expression do not have much impact on the paper.
>
> **Q5**: How did you preprocess the text? How did you preprocess the labels?
>
> **A5**: Regarding text preprocessing and labeling, we followed most of the previous work of IC-DST (https://github.com/Yushi-Hu/IC-DST) on preprocessing and labeling data. The only change is that we renamed some of the slot and domain names. The specific mapping rule dictionary is included in the self.kw2schema variable on line 26 of prompt_utils.py in the Supplementary Materials we submitted, which we hope will be helpful to you. We will consider explaining this in the appendix in future revisions.

---

### Official Review · Reviewer_9Gsy · 2023-08-07

**Soundness:** 4

**Excitement:**

3: Ambivalent: It has merits (e.g., it reports state-of-the-art results, the idea is nice), but there are key weaknesses (e.g., it describes incremental work), and it can significantly benefit from another round of revision. However, I won't object to accepting it if my co-reviewers champion it.

**Paper Topic And Main Contributions:**

This paper proposes ParsingDST, a new In context learning method that translates the original dialogue to json representation as an intermediate state.  Experiment results shows this method outperformance current SOTA on zero shot DST.

**Reasons To Accept:**

•	Results are presented well with detailed ablation study, The ablation analysis and case studies provide additional insight into the effectiveness of the approach.
•	It presents a novel approach with strong performance, and clear motivation.
•	This is a very well written paper, coherent and easy to understand.
•	Architecture of the model is well-defined in figure 2.
•	Overall, this is an effective paper with strong results.


**Reasons To Reject:**

Code can be released for other users to work on top of the current work

**Reproducibility:**

3: Could reproduce the results with some difficulty. The settings of parameters are underspecified or subjectively determined; the training/evaluation data are not widely available.

**Reviewer Confidence:**

4: Quite sure. I tried to check the important points carefully. It's unlikely, though conceivable, that I missed something that should affect my ratings.

---

> ### Author Rebuttal · Authors · 2023-08-24
>
> Thank you for your insightful suggestions and questions. Below is our detailed response:
>
> **Q**: Code can be released for other users to work on top of the current work.
>
> **A**: Thank you for acknowledging our work. As we stated in section 3.1 of our paper, we will release our complete code upon acceptance. We have also provided Supplementary Materials for reference, which include the main content of our code. We hope this will be helpful to you.

---

### Meta-Review · Area_Chair_tegW · 2023-09-08

**Recommendation:** 3

**Metareview:**

This paper studies the zero-shot dialogue state training problem. The authors introduce a novel in-context learning approach, ParsingDST, which utilizes large language models. This method translates DST into JSON via semantic parsing, acting as an intermediary state. The tests indicate that the proposed approach sets a new state-of-the-art for zero-shot DST settings.

The soundness scores for this paper stand at (4, 3, 2). In the initial submission, a prevalent concern among reviewers was the outdated evaluation benchmarks. However, the authors addressed this in their rebuttal by presenting results from the most recent benchmark. Consequently, there's a consensus among reviewers on the soundness of the experiments. The score of 2 for soundness primarily reflects the paper's methodological section, which could benefit from clearer exposition. We advise the authors to enhance the clarity of the technical details.

Regarding excitement, the scores are uniformly (3, 3, 3), indicating unanimous agreement among reviewers on this metric.

---

### Decision · Program_Chairs · 2023-10-07

**Decision:**

Accept-Findings

**Comment:**

This paper studies the zero-shot dialogue state training problem. The authors introduce a novel in-context learning approach, ParsingDST, which utilizes large language models. This method translates DST into JSON via semantic parsing, acting as an intermediary state. The tests indicate that the proposed approach sets a new state-of-the-art for zero-shot DST settings.

The soundness scores for this paper stand at (4, 3, 2). In the initial submission, a prevalent concern among reviewers was the outdated evaluation benchmarks. However, the authors addressed this in their rebuttal by presenting results from the most recent benchmark. Consequently, there's a consensus among reviewers on the soundness of the experiments. The score of 2 for soundness primarily reflects the paper's methodological section, which could benefit from clearer exposition. We advise the authors to enhance the clarity of the technical details.

Regarding excitement, the scores are uniformly (3, 3, 3), indicating unanimous agreement among reviewers on this metric.